# A theory for colors of strongly correlated electronic systems

Swagata Acharya [1,2] ✉, Dimitar Pashov [3], Cedric Weber[4],
Mark van Schilfgaarde[2], Alexander I. Lichtenstein [5,6] & Mikhail I. Katsnelson [1]

Many strongly correlated transition metal insulators are colored, even though they have band gaps much larger than the highest energy photons from the visible light. An adequate explanation for the color requires a theoretical approach able to compute subgap excitons in periodic crystals, reliably and without free parameters—a formidable challenge. The literature often fails to disentangle two important factors: what makes excitons form and what makes them optically bright. We pick two archetypal cases as examples: NiO with green color and $MnF_2$ with pink color, and employ two kinds of ab initio many body Green's function theories; the first, a perturbative theory based on low-order extensions of the $GW$ approximation, is able to explain the color in NiO, while the same theory is unable to explain why $MnF_2$ is pink. We show its color originates from higher order spin-flip transitions that modify the optical response, which is contained in dynamical mean-field theory (DMFT). We show that symmetry lowering mechanisms may determine how 'bright' these excitons are, but they are not fundamental to their existence.

Understanding how light interacts with matter poses an extraordinary challenge. The color of a compound is complementary to the wavelengths it absorbs; thus an adequate theory of the frequency-dependent dielectric response should be able to explain the color of real materials. Often the dielectric response is adequately described in terms of an independent particle (Lindhard) description. Graphite and diamond are respectively black and transparent, because the former has no bandgap and absorbs all wavelengths from the visible spectrum, while the bandgap of diamond lies outside this range. The colors of noble metals Cu, Ag, and Au ($d^{10}s^1$) are determined primarily by the wavelengths at which photons are reflected, which is controlled by the $d^{10}s^1 \rightarrow d^9s^2$ transition. Cu has the shallowest $d$ levels, which gives it the copper color, while they are much deeper in Ag and it appears silver. Au is intermediate between the two. Information for $d^n \rightarrow d^{n+1}$ transitions are encoded in one-particle Green's functions, and in such cases the absorption should be well characterized, provided the one-particle

Green's function is adequate and the electron-hole attraction is not strong. Colors of natural pigments, found in flowers and vegetables, have been explained[1–3] using time-dependent density functional theory[4–6] (TDDFT) and many body perturbative approaches[7,8]. For relatively weakly correlated systems and molecules[9–11] such approaches are sufficient to well reproduce the optical spectrum. Further, in some rare-earth based fluorosulphide pigments[12,13] the colors have been explained recently within a dynamical mean field theory (DMFT) framework. As for the other systems just mentioned, DMFT improves one-particle Green's function, adding higher order local spin fluctuation diagrams missing in many body perturbative GW based approaches. Such corrections are important in Hund's metals, Kondo-systems, $f$-electron systems, weakly doped Mott insulators.

However, it is well known that the one-particle Green's function may not be enough to describe the dielectric response. The vertex in the polarizability is a two-particle object responsible for electron-hole attraction, which gives rise, e.g., to excitons and optical

[1]Institute for Molecules and Materials, Radboud University, Nijmegen 6525 AJ, The Netherlands. [2]National Renewable Energy Laboratory, Golden 80401 CO, USA. [3]Theory and Simulation of Condensed Matter, King's College London, The Strand, London WC2R2LS, UK. [4]Quantum Brilliance Pty, The Australian National University, Gould Building (116), Daley Road, Canberra ACT 2600, Australia. [5]Institute of Theoretical Physics, University of Hamburg, Hamburg 20355, Germany. [6]European X-Ray Free-Electron Laser Facility, Holzkoppel 4, Schenefeld 22869, Germany. ✉e-mail: swagata.acharya@nrel.gov

absorption below the fundamental gap[14–20]. When the excitons are weakly bound (Wannier type) they involve mostly states near the edge of the fundamental gap: they are confined to a small volume of $k$ space and thus are spread over many lattice sites. This is the usual situation in most transition metal dichalcogenides[21–23]. The opposite (atomic or Frenkel) limit occurs when the collective excitation takes place between highly localized states, typically transitions between different configurations of a particular $\ell$ on an atom, e.g., a $d^n \to d^n$ transition. These atomic-like excitons are governed mostly by properties of the atom and depend only weakly on the environment. For this reason, they have been traditionally tackled via ligand field theory: an atom with a few ligands are attached to approximate the solid, and a relatively high-level quantum-chemical calculation performed. Atomic transitions that violate the $\Delta\ell = \pm 1$ selection rule are dipole forbidden (Laporte rule), and require some symmetry-breaking mechanism to occur, such as $p$-$d$ hybridization, spin-orbit coupling, phonons, and Jan-Teller distortions. Ligand-field theory has been a successful traditional line of attack, but it suffers from two difficulties. First, the brightness of the transition depends on the details of symmetry-breaking mechanism, which can be different in the solid than in the molecule. $CrI_3$ is a prime example: the 2D and bulk compounds have similar exciton energies but very different brightness. Moreover, ligand field theory cannot adequately capture excitons larger than the cluster size, and the artificial confinement affects the energy. This occurs when excitons are not entirely Frenkel-like. (Below we present an exciton in NiO as one instance of this.) In such situations, it is crucial to build a parameter-free ab initio theory able to reliably compute both one- and two-particle properties in the thermodynamic limit, to give a systematic understanding of what originates these excitons and what makes them bright.

$CrX_3$ is one system where excitons have recently been well characterized by low-order many-body perturbation theory[24–26]. Adding ladder diagrams to the RPA polarizability via a Bethe-Salpeter equation (BSE) in the particle-hole charge channel, yields the deep Frenkel excitons observed. To accomplish this, the one-particle Green's function must be also of high fidelity. In refs. 24,26 the LDA Green's function was augmented by a semiempirical Hubbard parameter, which yields improved energy levels but a somewhat inadequate description of the eigenfunctions; in ref. 25 a fully ab initio approach was employed. The latter technique, which we also use in the present work, extends the $GW$ approximation to include ladder diagrams in $W$ for the self-energy. Excitonic levels are similar in the two approaches, but the better eigenfunctions in the ab initio approach give rise to some differences[27].

$CrX_3$ is fully spin-polarized ($t_{2g\uparrow}^3$ with 3 $\mu_B$/Cr atom) ferromagnetic insulator. The valence and conduction band edges are $3d$-$t_{2g\uparrow}$ and $e_{g\uparrow}$ respectively which can host trivially triplet electron-hole excitations between $t_{2g}$ and $e_g$ states still keeping the atomic $d^3$ configuration. These heavily bound Frenkel excitons originate from a $d^3 \to d^3$ transition and should be dark according to the Laporte rule, but they became bright in periodic crystals, primarily because the valence and conduction $3d$ states strongly hybridize with X-$p$ states[25]. Spin-orbit coupling[28], odd-parity phonons, Jan-Teller distortions[29], and as we show here, spin disorder, can also lower the symmetries of the exciton wavefunctions can make them still more bright.

All the excitons in $CrX_3$ can be picked up by a $GW$+BSE approach since spin fluctuations are not involved. However, this not always the case, especially for antiferromagnetic insulators. The insulating band gap of NiO is ~4 eV[30]. However, NiO is pale green, which implies that at least a portion of the light-matter interaction originates from excitons that absorb selectively in the visible part of the optical spectrum (380–700 nm or 3.26–1.65 eV). Furthermore, NiO appears green in its bulk crystalline variant, in thin film and in powdered form, suggesting the presence of the deep-lying excitonic absorption common to all forms. An even more dramatic situation emerges in Mott insulating

$MnF_2$, where the one-particle band gap is ~8 eV[31–33], nevertheless, the material appears pink in its bulk crystalline variant. In NiO, Ni assumes a $d^8$ atomic configuration ($t_{2g}^6 d_{z^2}^1 d_{x^2-y^2}^1$) in the solid, while Mn is $d^5$ in $MnF_2$, all electrons having the same spin (Fig. 1). In analogy with $CrX_3$, in NiO, the Frenkel excitons with atomic $d^8 \to d^8$ transitions should still be picked up in many-body perturbative framework since an essentially atomic triplet exciton can emerge without requiring a spin flip. This is accomplished by creating a hole in the $t_{2g}$ orbitals and adding an electron in the half-filled $e_g$ orbitals, which can preserve the $<S>=1$ ground state atomic configuration of Nickel (Fig. 1). A prior theoretical work[34] did study excitons above 3 eV, but such excitons do not explain the color in NiO. This theory[34] constructs the dielectric function from GGA+U+$\Delta$ hamiltonian, Here $U$ is a Hubbard parameter and $\Delta$ is a scissors shift. $U$ and $\Delta$ are chosen to reproduce the best agreement against observed one-particle properties in NiO. Further, they produce good agreement against experimental macroscopic dielectric response by normalizing their theoretically computed response by the experimental dielectric constant. In fully antiferromagnetically ordered NiO such triplet transitions should be dark according to the Laporte rule; however, as we have already discussed, they nevertheless exist, and moreover can become bright owing to a symmetry-lowering mechanism, as we will discuss below. The situation is completely different in $MnF_2$, where any atomic electron-hole transition, $d^5 \to d^5$, necessarily involves a spin-flip mechanism (Fig. 1). Such excitons are completely absent from a standard $GW$+BSE framework and needs a higher-level theory that incorporates local spin fluctuations that modify the charge component of the two-particle Green's function.

We will employ two approaches: the $GW$+BSE perturbative approach and an augmentation of $GW$ with dynamical mean field theory (DMFT). In all cases the starting Hamiltonian that generates the response function is computed self-consistently from $GW$ itself; this is essential to to achieve high fidelity. A brief description of both can be found in the Methods section below; see also refs. 35–37 for a description of MBPT, and refs. 36,38 for DMFT. MBPT includes non-local charge correlations but misses out on spin fluctuation diagrams, while in DMFT nonlocality is limited to orbitals on an atomic site. Within that restriction, it is locally exact and incorporates all local spin fluctuation diagrams. We will establish that the one-particle properties of both NiO and $MnF_2$ are equally well described by either approach. However, the perturbative approach fails to explain excitons in $MnF_2$ and needs the locally exact method with spin fluctuation diagrams to yield its pink color. In the process, we will discuss the crucial technical ingredients of such theories and its ability in predicting collective responses in strongly correlated systems. We will also show that our theory can compute reliably the eigenvalues and eigenfunctions of all such fundamentally dark excitons in anti-ferromagnetic insulators, however, symmetry-lowering-mechanisms that are present in a real materials are crucial in determining the 'brightness' of these excitons. Using DMFT, we will explore the scaling behavior of the eigenvalues of these excitons with Hund's coupling. We will establish that the hardest problem is to explain the interactions that are responsible for the pink color in $MnF_2$, since it is a strictly half-filled antiferromagnet. However, we are able to solve the problem in its entirety and will establish that spin-flip vertex that knows about the physical spin-disordering mechanism of the atomic multiplet structure is responsible for its color.

## Results and discussion
### Many-body perturbative theory: 'dark' excitons in the visible range in antiferromagnetic NiO
We begin with a study of NiO, and employ a many-body perturbative Green's function $GW$ theory QS$G\widehat{W}$[18,25,37,39], an extension of the quasiparticle self-consistent $GW$ approximation (QS$GW$)[35,36,40], where the polarizability needed to construct $W$ is computed including vertex corrections (ladder diagrams) by solving a Bethe-Salpeter equation

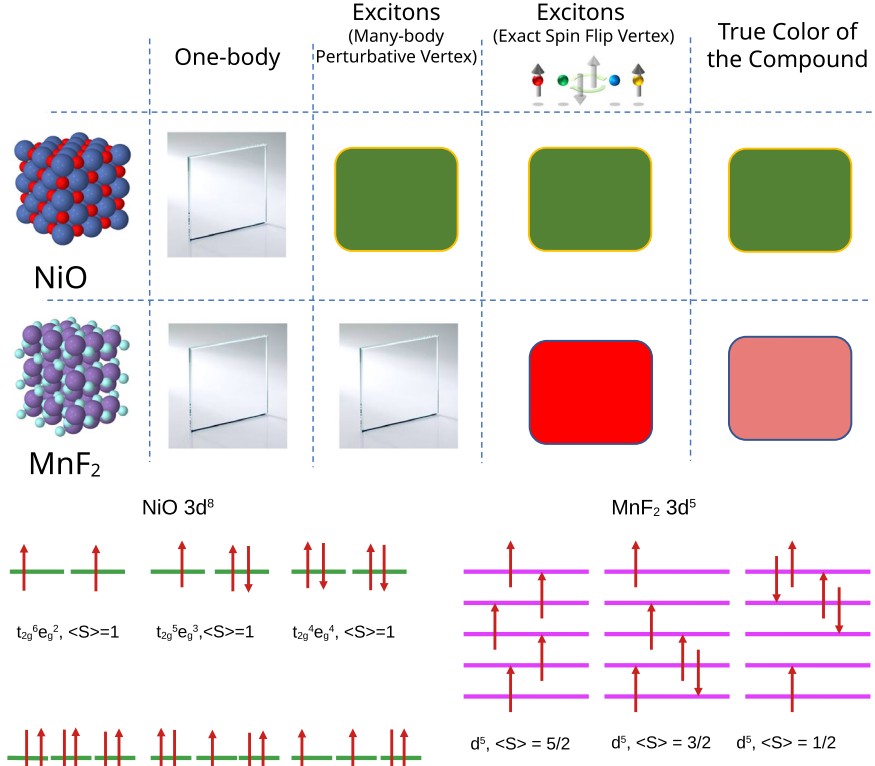

**Fig. 1 | Fundamentals of the colors in NiO, MnF₂.** Schematic depiction of fundamental atomic collective charge excitations in NiO and MnF₂. In NiO $d^8$ to $d^8$ two-particle charge transitions are possible between $t_{2g}$ and $e_g$ states that preserve the atomic $<S>=1$ spin configuration. However, in MnF₂ any such collective atomic charge excitations are associated with a simultaneous change in the atomic spin configuration. In NiO, such fundamentally triplet charge excitations determine the green color of the material, while in MnF₂ charge excitations associated with a change in atomic spin configuration determines its pink color. NiO appears green when electron-hole vertex is treated within a paramagnetic formulation of the theory, both nonlocal low-order many-body perturbative and locally exact high-order theory. MnF₂ remains transparent within the paramagnetic many-body perturbative theory, and becomes colored only when the spin-flip component of the electron-hole vertex is incorporated.

(BSE) for the two-particle Hamiltonian[37]. There are some crucial differences in the implementation of QS$G\widehat{W}$ presented here from other widely used implementations of BSE. Self-consistency in both Σ and the charge density[39,41] plays a crucial role: the theory becomes uniformly consistent so that uncontrolled ad hoc empirical additions are not needed. Computing the screened coulomb interaction $\widehat{W}(\omega,q)$ with ladders included in the polarizability largely eliminates the tendency for QS$GW$ to overestimate the fundamental bandgap. (The vertex in the construction of $\widehat{W}$ is approximated with a static RPA $W$, as is customary[21–23]). $G$, Σ and $\widehat{W}$ are iterated until all of them converge. In each cycle, $W$, is made anew, and the four-point polarizability is computed to make $\widehat{W}$ from an updated $W$.

With a self-consistent self-energy $\Sigma = iG\widehat{W}$ in hand, we use the BSE to compute the macroscopic dielectric response function in NiO, initially modeling the antiferromagnetically (AFM) ordered phase. Within QS$G\widehat{W}$, NiO has 4.0 eV fundamental band gap compared to 5.0 eV (Supplementary Fig. 1) from QS$GW$[37,42]. This shows that the vertex significantly modifies the self-energy in NiO[18,39], softening $W$ and thus reducing the gap. The seminal work by Sawatzky and Allen[30] estimated the gap to be ~4.3 eV from Bremsstrahlung inverse spectroscopy (BIS). Our QS$G\widehat{W}$ one-particle spectra agrees remarkably well with the BIS data[37]. A recent XPS study caps the gap at 4.0 eV (within 0.35 eV energy resolution)[43]. Next, we compute the vertex corrected macroscopic dielectric response functions and find that there are in-gap optical absorption down to 3.6 eV, i.e., $E_b$ ~ 0.4 eV deeper than the edge of the one-particle band gap in QS$G\widehat{W}$. Our computed imaginary part of macroscopic dielectric response $\epsilon_2$ agrees remarkably well[37] with the experimental data from Powell and Spicer[44] over the full frequency range. This shows with a suitably vertex corrected self-energy, QS$G\widehat{W}$

incorporates the relevant electron-hole vertex corrections that can red-shift the optical spectral weight by a significant amount. A previous semi-empirical GW+BSE calculation[34] shows similar optical absorption inside band gap around 3.6 eV. Nevertheless, such optical absorption does not produce the right color. Moreover, without phenomenological adjustments such as was done in ref. 34, single-shot calculations often have limited success in describing the optical absorption in antiferromagnets. Nor is it sufficient to well describe the one-particle spectrum: the electronic eigenfunctions are biased by the density-functional eigenfunctions and yield poor description of their orbital character. We discuss this aspect in detail in the supplemental materials and show explicitly how our self-consistent vertex corrected approach systematically corrects the electronic eigenfunctions in a fully diagrammatic fashion without the need of ad hoc parameters.

We find strong optical absorption starting at 3.6 eV, as we have discussed elsewhere[37]. We also find a deep-lying eigenvalue of the two-particle e-h Hamiltonian at -1.6 eV, but the oscillator strength of this eigenvalue is five orders of magnitude smaller than the optical shoulder at 3.6 eV (Fig. 2). This exciton has the right energy to absorb the red part of the visible spectrum, thereby causing NiO to appear green. We resolve the corresponding eigenfunction in the atomic, orbital, band basis and also in real space (Fig. 2). Nearly 70% of the spectral weight comes from on-site and ~20% comes from inter-site $e_g$-$t_{2g}$ transitions. The rest of the spectral weight comes from e-h processes shared between Ni and O. The situation is dramatically different for absorption from the 3.6 eV exciton. The inter-site $e_g$-$t_{2g}$ transition becomes the most dominant mechanism followed by processes where electron and holes are shared between Ni and O. The Ni on-site element almost entirely vanishes at optical shoulder, and the oscillator

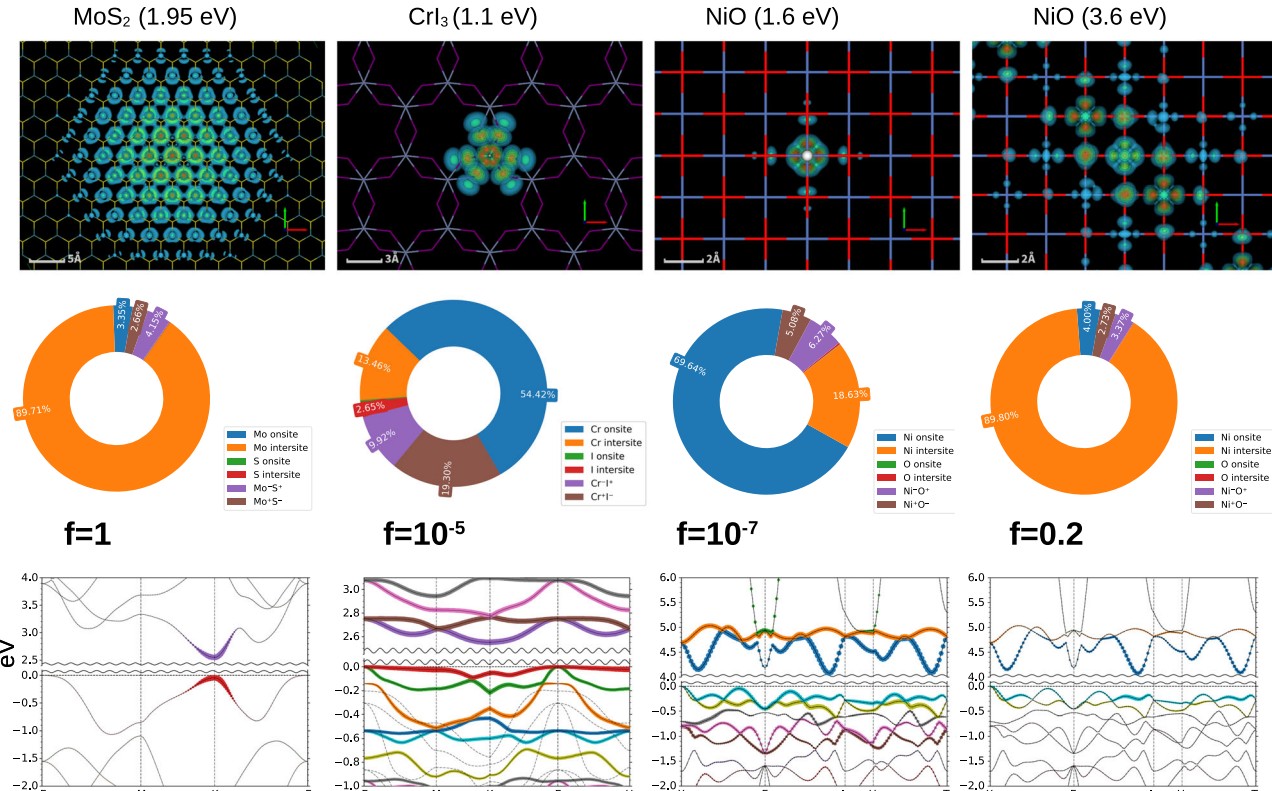

**Fig. 2 | Orbital, atomic, momentum and real-space decomposition of the excitons.** The deepest lying excitons are shown for non-magnetic $MoS_2$, ferromagnetic $CrI_3$ and AFM-NiO. The exciton at 1.95 eV (binding energy $E_b = 0.55$ eV) in $MoS_2$ extends to several nanometers while the excitons at 1.6 eV ($E_b = 2.4$ eV) in NiO and at 1.1 eV ($E_b = 1.4$ eV) in $CrI_3$, are localized to ~4 Å only. Bands that are colored take part in the exciton formation, while the width of the bands signify relative contributions to the excitonic eigenfunctions. We use different colors to identify different bands that take part in the exciton formation. The curly line cuts are used to represent the energy discontinuity between the valence band top and conduction band bottom. Several valence and conduction bands and all electron and hole momenta contribute almost uniformly to the formation of the Frenkel excitons (at 1.1 eV in $CrI_3$ and 1.6 eV in NiO) while the weakly bound Wannier-Mott excitons (at 1.95 eV in $MoS_2$ and at 3.6 eV in NiO) are mostly formed from holes at valence band top and electrons at conduction band bottom. The Frenkel excitons are dominantly on-site $d$-$d$ in nature, with sub-leading inter-site $d$-$d$ contributions. However, the exciton at the optical shoulder is almost entirely inter-site $d$-$d$ in nature. The oscillator strengths ($f$) of the Wannier-Mott excitons are at least four-five orders of magnitude higher than the Frenkel excitons. The 1.6 eV exciton in NiO is darker compared to the exciton at 1.1 eV in $CrI_3$ because it has more on-site $d$-$d$ component and less $p$-$d$ component.

strength, as a consequence, is ~5 orders of magnitude larger than the 1.6 eV exciton.

To make this exciton bright, some symmetry-lowering mechanism is needed. Spin-orbit coupling (SOC) is one obvious candidate; indeed the classic works by Sugano and Tanabe[28,45] used a combination of atomic ligand field theory and SOC as the main theoretical foundation that relaxes atomic Laporte rule to explain excitonic absorption in several transition metal oxide insulators. We find that SOC, indeed, enhances the oscillator strength of the 1.6 eV absorption by three orders of magnitude (Supplementary Table 1), but it still remains 'dark' enough that it can not be observed in absorption spectra (Supplementary Fig. 5). To put it in perspective, its oscillator strength (even after including SOC) remains at least two orders of magnitude smaller than the darkest ground state exciton from $CrI_3$. Also, this is completely consistent with the optical absorption data reported by Powell and Spicer[44]. They don't see any peaks in their absorption data below the 3.6 eV shoulder, while Propach and Reinen[46] do see at least three peaks below the 3.6 eV in their optical emission spectra. The possible reasons for how these deep lying dark excitons can brighten in the emission spectroscopy but not in the absorption, will be discussed later. Also in a later section why SOC can not be the mechanism either for the presence or for the brightening of these dark excitons will be discussed in detail. However, other mechanisms can enhance the brightness of dark excitons as well. In extended systems there is a finite $d$-$p$ hybridization, as we showed recently for two-dimensional

magnetic semiconductors $CrX_3$[25,27]. Contrasting NiO and $CrX_3$ sheds light on their differences: the deepest lying bright exciton (at 1.1 eV) in ferromagnetic monolayer of $CrI_3$ has two orders of magnitude higher oscillator strength compared to the 1.6 eV exciton in AFM NiO. As Fig. 2 shows, the excitonic spectral weight in $CrI_3$ has nearly 30% contribution coming from processes where electron and holes are shared between Cr and I, in strong contrast to 1.6 eV exciton in NiO that has only about 10% contribution from such processes. Both of them are fundamentally triplet and mostly on-site in nature. Computing them both in their crystalline environment without assumptions behind ligand-field theory makes it possible to compare the role of the lattice, and moreover show explicitly the quantitative nature of the difference in their spectral decomposition. This is a step change in the study of collective charge excitations in antiferromagnetic insulators. Having said that, we will show in the following sections the primary processes that are responsible for making these excitons 'brighter', without having to introduce ad hoc arguments for the symmetry lowering mechanism.

### Many-body perturbative theory: 'bright' excitons in the visible range in paramagnetic NiO

Spin disorder can be a mechanism to turn dark excitons bright. To show this we construct a Special QuasiRandom Structure[47], a $2 \times 2 \times 2$ supercell of the AFM structure consisting of 16 Ni in a (pseudo) paramagnetic spin arrangement and compute the self-energy with

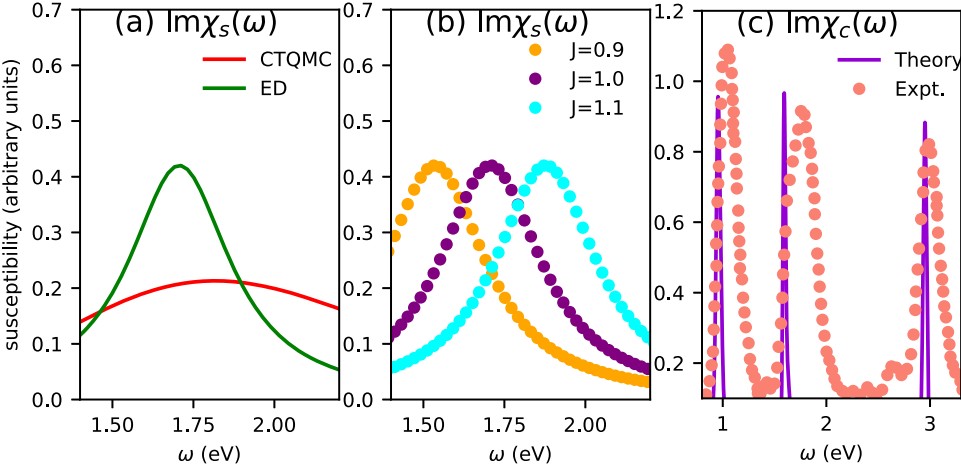

**Fig. 3 | Peaks in spin and charge susceptibilities at different energies from an exact many-body approach. a** Imaginary part of dynamic spin susceptibility Im$\chi_s(\omega)$ computed in paramagnetic DMFT, in presence of local vertex corrections, show a collective spin excitation at 1.7 eV. The energy of the excitation is robust across choices of different impurity solvers (ED or CTQMC) in DMFT. This peak is associated with the triplet to singlet transition on the atomic site. **b** The energy of this peak scales with Hund's coupling $J$ as 1.7 $J$, suggesting absence of this spin

transition for $J = 0$. **c** Experimentally observed peaks in the collective excitations as measured by non-resonant inelastic X-ray scattering measurements[58], which are insensitive to optical matrix elements, are plotted alongside the charge susceptibility computed from DMFT. The intensity of the charge peaks in DMFT are multiplied by a large factor to bring them to same scale with experiment. The number and the energy of the peaks agree remarkably well with the experiment.

QS$G\widehat{W}$. The self-consistent PM and AFM local moments are very similar ($|M| = 1.76 \pm 0.01 \mu_B$ and $1.72 \mu_B$ in the two cases), and the PM band structure strongly resembles the AFM when folded to a $2 \times 2 \times 2$ supercell. The QS$G\widehat{W}$ (QS$GW$) gap (Supplementary Fig. 3), differs from the AFM gap by 0.2 eV. This flatly contradicts the commonly made assertion[48] that NiO cannot be described in terms of a conventional Slater (band) theory, which "attribute[s] the insulating gap to the existence of long-range magnetic order[49]", and requires a dynamical theory such as DMFT for an adequate description of the electronic structure. The input for a DMFT calculation is usually a non-magnetic metallic phase, a charge (Mott) gap opens because of dynamically fluctuating local moments, according to the theory. The present calculations establish conclusively that local magnetic moments are essential to form the gap, but that dynamical fluctuations are not. A similar conclusion was reached in a recent work on Mott insulators[50]. Thus, the formation of the gap can be equally represented by a frequency-dependent site-local potential such as used in DMFT, or a low-order perturbative approach like $GW$. The latter has important advantages: it is fully ab initio and simpler, without the complications of partitioning into a correlated subspace with a parameterized Hamiltonian; and it well describes optical excitations over a wide energy window. The single-most important contribution of spin-disordering is that it makes the 1.6 eV exciton bright, enhancing its oscillator strength by at least three orders of magnitude compared to the fully ordered ideal AFM situation. The matrix elements between up- and down-spins do not vanish in the paramagnetic state, making this peak optically bright (Fig. 3). While the fully disordered limit is applicable only above the Néel temperature, spin fluctuations occur at all temperatures[51]: thus they can be an important contributor to the brightness of the 1.6 eV exciton. In a future work, we will assess the relative importance of this effect and spin orbit coupling on the color in NiO.

**Many body locally exact theory: excitons in the visible range from paramagnetic DMFT in NiO**
In a parallel approach, we perform a conventional non-magnetic QS$GW$ +paramagnetic DMFT[52]. The impurity Hamiltonian is built out of the Ni-3$d$ orbitals, which hybridize with the bath. All five Ni-3$d$ orbitals which are comprised of $t_{2g}$ and $e_g$ states, are kept in the Hubbard

Hamiltonian. The Anderson impurity model is solved in the presence of a rotationally invariant interaction matrix. We find that the Mott-Hubbard gap opens through DMFT self-consistency and the band gap is ~ 4 eV for U = 8 eV and $J = 1.0$[53]. We solve the Anderson impurity model with two different exact impurity solvers, exact-diagonalisation (ED) and continuous-time Quantum Monte Carlo (CTQMC)[54,55], and find that all the essential conclusions are independent of the choice of the solver. We compute the local spin ($\chi_s$) and charge ($\chi_c$) susceptibilities on real frequencies from ED and we resolve them in their intra- and inter-orbital components. In Fig. 3(a) we show that the energy of the peak in $\chi_s$ is essentially same from CTQMC and ED. Within paramagnetic DMFT, the magnetic vertex can be written as $\Gamma_s = \Gamma_{\uparrow\uparrow} - \Gamma_{\uparrow\downarrow}$ and the charge vertex as $\Gamma_c = \Gamma_{\uparrow\uparrow} + \Gamma_{\uparrow\downarrow}$. It is only natural that spin and charge susceptibility peaks appear at different energies in paramagnetic DMFT. DMFT local irreducible vertex functions are site-local, in contrast to the non-local static vertex in QS$G\widehat{W}$. The most remarkable difference between the two is the spin-flip component of the vertex in DMFT, $\Gamma_{\uparrow\downarrow}$, which is completely absent within QS$G\widehat{W}$.

Three distinct peaks are observed in Im$\chi_c$ in the energy window 0–3 eV: at 0.93 eV, 1.55 eV and 2.91 eV (Fig. 3c). All these two-particle charge excitations preserve the $<S> = 1$ atomic configuration on average (Fig. 1). The three peaks correspond to $t_{2g} \rightarrow e_g$ transitions: excitations of the ground state $t_{2g}^6 e_g^2 \rightarrow t_{2g}^5 e_g^3$, $t_{2g}^6 e_g^2 \rightarrow t_{2g}^4 e_g^4$ and a third which is combination of the first two. The peaks found when ED is used to solve the Anderson impurity problem are also found when the CTQMC solver is used; but as the CTQMC bath is not discretised the peaks are broadened and harder to detect. Nevertheless, the three-peak structure is robust and is independent of the DMFT solver. Intriguingly enough, two of these three peak positions match almost exactly with the excitons observed in paramagnetic QS$G\widehat{W}$ (Fig. 4).

**Distinct scaling features for peaks in spin and charge susceptibilities with Hund's coupling**
The spin susceptibility $\chi_s$ is somewhat different. Irrespective of whether only the (half-filled) $e_g$ states or $t_{2g}$ and $e_g$ states are all included in the impurity Hamiltonian, Im$\chi_s$ always shows a single high intensity absorption peak at 1.7 eV (Fig. 3a). In the spin channel, this peak is purely associated with a triplet-singlet

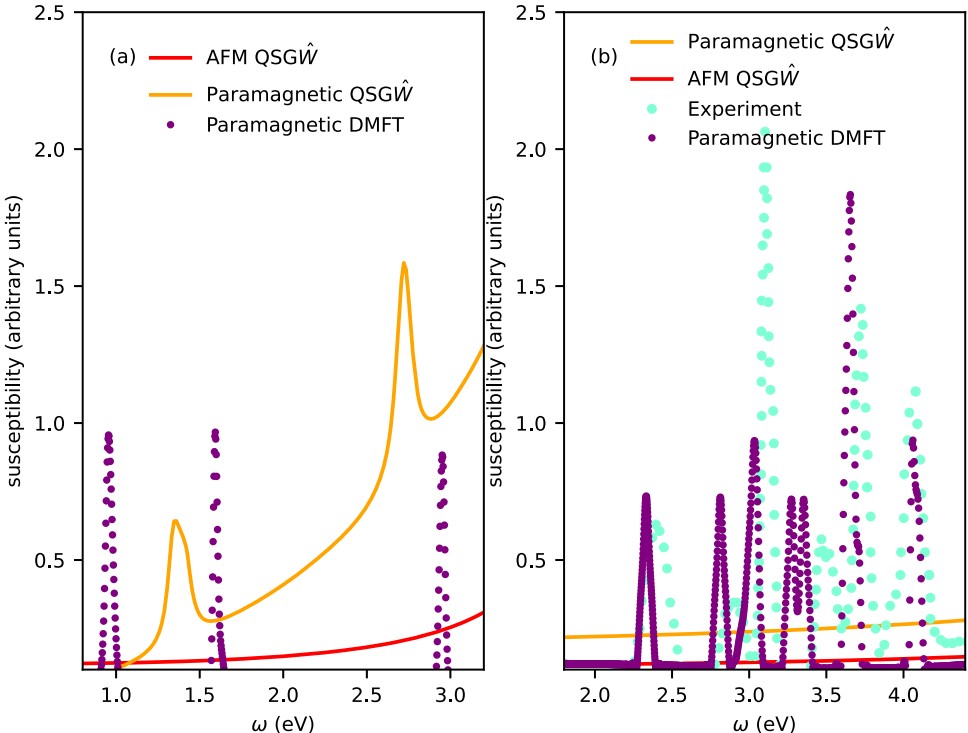

**Fig. 4 | Colors in NiO and MnF₂ determined by the essential structure of the electron-hole vertex from perturbative and exact many-body approaches.**
**a** The imaginary part of the charge susceptibility $\chi_c$ computed from different levels of the theory and from experiment. In NiO, both paramagnetic QS$G\hat{W}$ and paramagnetic DMFT can produce the right color of the material as they account for the peaks at ~1.6 eV and ~3 eV that essentially absorbs the red and blue part of the visible spectrum respectively and can make NiO appear green. The missing peak from paramagnetic QS$G\hat{W}$, compared to DMFT, at ~1 eV is outside the visible range and does not contribute to the color of the material. **b** In MnF₂ it is the spin flip component of the DMFT charge vertex that is essential to the collective electron-hole charge excitation on the atomic site of Mn. So the collective excitations are only observed in DMFT and are completely absent from both AFM QS$G\hat{W}$ and paramagnetic QS$G\hat{W}$. The charge susceptibility peaks observed in DMFT agree remarkably well with the series of optical absorption peaks observed in experiments[28,60].

($<S> = 1 \rightarrow <S> = 0$) transition which changes the ground state from $t_{2g}^6 d_{z^2}^1 d_{x^2-y^2}^1$ to $t_{2g}^6 d_{z^2}^0 d_{x^2-y^2}^2$ or $t_{2g}^6 d_{z^2}^2 d_{x^2-y^2}^0$, as both of them are equally probable due to the cubic crystal field). The 1.7 eV peak scales with Hund's coupling $J$ in a manner completely different from the peak in Im$\chi_c$ at a similar energy (Fig. 3b) and Table 1 suggesting that a coupling between them mediated by SOC that would make this exciton bright would be sufficient but not fundamentally necessary. In a pure atomic multiplet transition ($<S> = 1$ to $<S> = 0$) the peak should be at $2J$, but it is at $1.7J$ and scales as $1.7J$, as determined by doing calculations varying $J$ as a parameter; see Fig. 3b). This suggests that a purely atomic scenario to explain the fundamental optical and magnetic transitions is only partially valid:, additional spin and charge fluctuations are present, which are correctly incorporated within a paramagnetic calculation. The optical

**Table 1 | Distinct scaling features of the peak energies in charge and spin susceptibility Im$\chi_c$ and Im$\chi_s$ with Hund's coupling $J$**

| Theory | Charge peaks (eV) | | | Spin peaks (eV) |
|---|---|---|---|---|
| $J = 0.8$ eV | 0.93 | 1.52 | 2.6 | 1.36 |
| $J = 0.9$ eV | 0.93 | 1.535 | 2.76 | 1.53 |
| $J = 1.0$ eV | 0.93 | 1.55 | 2.91 | 1.7 |
| $J = 1.1$ eV | 0.93 | 1.57 | 3.06 | 1.87 |

The peak labeled "spin peak" is computed from Im$\chi_s$ and scales as $1.7J$. Peaks in Im$\chi_c$ (labeled "charge peak") scale differently with $J$. In particular, the exciton peak at 1.6 eV (essential for the green color in NiO), is insensitive to $J$, suggesting no cross coupling with $\chi_s$, though a peak is present in both $\chi_s$ and $\chi_c$ at a similar wavelength.

spectrum computed from the paramagnetic phase either in QS$G\hat{W}$ and QS$GW$+DMFT, produce the long-sought green color of NiO. The low energy peak at ~1.6 eV partially absorbs the red part of the visible spectrum, while the absorption above ~2.8 eV (Fig. 4a) partially absorbs the blue part of the visible spectrum, leading to the green color of NiO (Fig. 1). Note that the peaks in Im$\chi_c$ from DMFT are low intensity as they are dipole-forbidden. Such excitations are entirely absent in a purely one-particle picture. One-particle Green's functions allow excitations of the atomic states of the form $3d^8 \rightarrow 3d^7$ or $3d^8 \rightarrow 3d^9$ (one electron on the atomic site goes into the bath or arrives from the bath).

These low-energy peaks emerge from the local vertex in the two-particle Green's function that allows $3d^8 \rightarrow 3d^8$ transitions. These peaks can become bright if any symmetry breaking mechanism such as spin-orbit coupling is present. However, as we discuss above, SOC is not fundamental to these transitions, nor to the essential charge absorption that determines the color. Our observations are fully supported by recent X-ray measurements[56–58], which is insensitive to the optical matrix elements, where all these three peaks are observed at almost the same energies as our theory (Fig. 3c). Intriguingly enough, in the classic experimental work by Propach and Reinen[46], they picked up all the three peaks in the optical emission spectra. Powell and Spicer[44], however, failed to see these three peaks in their optical absorption spectra. This is only natural since e-h emission processes can couple to non-radiative mechanisms present in the crystal and that can make the emission lines bright enough to be picked up in spectroscopy. In essence, this re-affirms our theoretical results that these excitons primarily emerge from the essential structure of the two-particle electronic interactions (vertex) and it is their brightness that is determined

by spin-orbit coupling or coupling of these excitons to different bosons.

Further, linear scaling of the peak in $Im\chi_s$ also implies that as $J \to 0$, the peak in spin susceptibility must vanish. However, no such scaling holds for the charge peak at ~1.5 eV: its position changes little as $J$ is varied. This is another signature for the charge peaks, that distinguishes them from the spin. We find the 0.93 eV and 1.55 eV peaks to be independent of $J$ and the 2.9 eV peak to scale with $J$ as 1.40 eV + 1.5 $J$. All three charge absorption peaks should survive even when $J$ is 0. The ~1 eV peak seen in DMFT but absent from QS$G\widehat{W}$, originates primarily from to the dynamics in the vertex in DMFT, which was omitted from the QS$G\widehat{W}$ vertex. However, this low-energy peak does not contribute to the color of NiO as it is far outside the visible range.

**Many-body perturbative theory: absence of excitons in the visible range in both antiferromagnetic and paramagnetic MnF$_2$**
We now turn to MnF$_2$. The reduced symmetry modifies the crystal field: now the five Mn $d$ states are non-degenerate and each is half-filled (Fig. 1). This differs qualitatively from NiO, with its filled degenerate t$_{2g}$ and half-filled degenerate e$_g$ levels. The scenario for possible optical transitions is thus much wider in MnF$_2$, while on the other hand, any $d^5 \to d^5$ transition requires a spin flip. When AFM QS$GW$ is compared to paramagnetic QS$GW$, the one particle gaps are slightly different (9.1 eV for the AFM case and 8.6 eV for the PM case, QS$G\widehat{W}$ gap for AFM case is 8.4 eV), similar to NiO (Supplementary Fig. 2). These numbers are in the right ballpark with available experiments (the fundamental gap for MnF$_2$ is not very well known, with experimental reports ranging from 8–10 eV[31,32,59]). For the charge susceptibility the lowest eigenvalue of the two-particle e-h Hamiltonian found to occur at 6.2 eV, for both AFM and PM cases. For the present purposes, the essential point is that the lowest eigenvalue of the QS$G\widehat{W}$ two-particle Hamiltonian is outside the visible (6.2 eV) which predicts MnF$_2$ to be colorless, inconsistent with the faded pink color observed.

**Spin-flip vertex as the fundamental component for the excitons in the visible range in MnF$_2$**
The 6.2 eV excitation is dark in AFM QS$G\widehat{W}$ and bright in paramagnetic QS$G\widehat{W}$, similar to the 1.6 eV excitation in NiO. However, to explain the pink color of MnF$_2$ some absorption in the visible range is necessary, which is not captured by QS$G\widehat{W}$. It is, however, captured by DMFT: using non-magnetic QS$G\widehat{W}$+paramagnetic DMFT we find a series of excitonic peaks in $Im\chi_c$ that purely originate from the dynamic vertex in DMFT (Fig. 4). This is naturally understood from the atomic configuration of MnF$_2$; a local charge excitation is allowed only when a spin-flip component $\Gamma_{\uparrow\downarrow}$ is present in the charge vertex (see Fig. 1). This is why the charge excitations in the visible range occur only because a *high-order, dynamical vertex* can preserve the atomic spin configuration on average but at any moment in time it allows for transitions between different spin states (from $<S> = 5/2$ to $<S> = 3/2,1/2$). With these series of optical absorption we observe that MnF$_2$ appears light-red in color (Fig. 1). The theoretical peak energies agree remarkably well with the experimental optical absorption spectrum[28,60] in the visible range. The color generated from $\chi_c$ by DMFT is very close to pink (the exact color is very sensitive to the precise details of the intensities and positions of these peaks). Our observations are completely consistent with a prior two-band Hubbard model calculation[61]. There it was shown that for two half-filled Hubbard bands, a charge peak appears at ~2$J$. Within such a model the only allowed inter-orbital transition involves a spin-flip mechanism. Such optical transitions can occur via a spin-flip vertex, although such a $d \to d$ transition would be forbidden in an atom. This two-band model is free of any other candidate symmetry breaking mechanisms usually considered that relaxes the Laporte rule, and it seems to contain the essential principles in our detailed ab initio calculation, which well reproduces the observed absorption spectrum. The intensity of the observed peaks within

DMFT are arbitrarily multiplied by a large factor to bring them to the same scale as experimental peaks. Since the symmetry lowering mechanisms are absent from our approach, we can not account for the brightness of these peaks. To put it in perspective, the intensities of these peaks from DMFT approach, in both NiO and MnF$_2$, are at least 6–7 orders of magnitude weaker than the peaks in transition-metal dichalcogenides and 2–3 orders weaker than the peaks in CrI$_3$. However, as we have explained in the previous sections, there is no reason why intensities of our computed excitonic peaks in optical absorption should agree with the intensities of the peaks observed in IXS or optical emission spectroscopy.

Strongly correlated electronic systems with a large fundamental gap can exhibit colors only because of collective charge excitations deep within the gap, which information is contained in the vertex in the two-particle susceptibility. Understanding origins of these collective excitations in a specific material can pose a formidable challenge. For transition metal compounds such as NiO and MnF$_2$ studied here, these excitations originate from dipole-forbidden $d \to d$ transitions. By applying two complementary high fidelity, ab initio approaches to the vertex function, we are able to fully explain the observed subgap absorption spectra and the colors correlative to them. Both approaches, a perturbative approach with a non-local but low-order vertex, and a local vertex with all graphs included, can produce the desired collective charge excitations in NiO that determines its green color. However, the deep excitons in MnF$_2$ responsible for its pink color are not captured by low-order perturbation theory. The locally exact method succeeds because it is a nonperturbative theory, which contains high-order spin flip diagrams that also contribute to the charge vertex. Spin disorder has only a slight effect on the one-particle Green's function, and similarly on the exciton energy levels that emerge from the two-particle Green's function. We showed, however, spin disorder serves as a symmetry-breaking mechanism, one among several such as spin-orbit coupling, vibronic mechanisms, $pd$ hybridization, and defects, all of which affect the exciton's brightness by enhancing their oscillator strengths.

Since the normally large dipole matrix element is absent by symmetry, the relative strength of such symmetry-breaking mechanisms that give rise to smaller matrix elements are important in the determination of the exact color, since they control intensities of the peaks in the visible range. However, the presence and energetics of these excitonic states are largely independent of those mechanisms, including spin disorder, and only sensitive to the adequate physical component to the vertex function, screened coulomb exchange and its spin components, which we have in our theory. We believe, this is a step change in the study of collective charge excitations of strongly correlated systems in that we are able to disentangle key mechanisms responsible for the emergence of the excitons and mechanisms that control their brightness. In the process, we establish the viability and limitations of two of the most effective approaches in studying collective excitations ab initio; DMFT and GW+BSE.

Study of collective charge excitations has come to the forefront of interesting physical phenomena in diverse fields in recent years, with advances in resonant, non-resonant, elastic and in-elastic X-ray scattering measurements that are combined with atomic force microscopy and scanning tunneling microscopy. Further, understanding atomic-like excitons and their spin components have been at the forefront of the research in optoelectronics[27,62], single-photon emitters and even qubits[63–65]. We believe, our work provides a timely guide to this field. The fact that we can compute these excitonic eigenvalues and eigenfunctions without any subscription to the atomic ligand field theory, we believe, is a desired step in the right direction. What we believe we understand can change or lead to enhanced understanding only when we are able to compute observables explicitly. We show, being able to compute these excitons and their spin components, provide us with

that key understanding of these systems and the desired theoretical methods for them, in parallel to solving an long-standing problem of essentially many-body nature.

## Methods

### Simulations of the ordered anti-ferromagnetic phase: LDA, QS*GW* and QS*GŴ* self-consistency

For the simulations of the ordered AFM phase 4 and 6 atom unit cells of NiO and $MnF_2$ were used respectively. Single particle calculations (LDA, and energy band calculations with the static quasiparticlized QS*GW* self-energy $\Sigma^0(k)$) were performed for both NiO and $MnF_2$ on a $12 \times 12 \times 12$ $k$-mesh while the (relatively smooth) dynamical self-energy $\Sigma(k)$ was constructed using a $4 \times 4 \times 4$ $k$-mesh and $\Sigma^0(k)$ extracted from it. For each iteration in the QS*GW* self-consistency cycle, the charge density was made self-consistent. The QS*GW* cycle was iterated until the RMS change in $\Sigma^0$ reached $10^{-5}$ Ry. Thus the calculation was self-consistent in both $\Sigma^0(k)$ and the density. Numerous checks were made to verify that the self-consistent $\Sigma^0(k)$ was independent of starting point, for both QS*GW* and QS*GŴ* calculations; e.g. using LDA or Hartree-Fock self-energy as the initial self energy for QS*GW* and using LDA or QS*GW* as the initial self-energy for QS*GŴ*. In the two-particle Hamiltonian 16 valence and 16 conduction bands were incorporated to achieve convergence. The dielectric response is converged on a k-mesh of $4 \times 4 \times 4$.

### Simulations of the paramagnetic phase: QS*GW* and QS*GŴ* self-consistency

For the disordered paramagnetic phase the supercell size is 32 for NiO and 48 for $MnF_2$. The supercell paramagnetism is simulated using special quasi-random structure prescription of Alex Zunger[47]. Nearest neighbor, two-body nearest neighbor and three body nearest neighbor correlation functions are made in a way to match exactly that of a random alloy. For NiO and $MnF_2$ QS*GW* and QS*GŴ* calculations were performed on a $2 \times 2 \times 2$ k-mesh for the self energy. Solution of the two-particle Hamiltonian is converged with 96 valence and 96 conduction bands (Supplementary Fig. 4).

### Paramagnetic DMFT calculations using ED and CTQMC impurity solvers

For DMFT both ED and CTQMC solvers are used. CT-QMC calculations are performed on five $d$-orbitals of the $3d$ transitions metals. For NiO, while performing nonmagnetic QS*GW*+paramagnetic DMFT, the Hubbard model in DMFT is solved using U = 8 eV, J = 1.0 eV and double counting correction of 55 eV as proposed in a previous work[53]. This allows us to produce exactly 4 eV of one-particle gap in paramagnetic DMFT. Once the one-particle experimental gap is reproduced, we explore the local vertex corrected dynamic susceptibilities in charge and spin channels inside the one-particle gap. For $MnF_2$, the Hubbard model on the five Mn $3d$ orbitals is solved within DMFT using U = 16 eV, J = 1.3 eV and double counting correction of 68 eV. Note that in both the cases the used double counting correction is slightly smaller than the fully localized limit, for the reasons explained in detail in this earlier work[53]. When ED is used as the impurity solver, the bath parameters are chosen in a way that reproduces the fully self-consistent CTQMC +DMFT self energy. For ED, both the Lanczos and exact brute force ED calculations are performed. The essential collective charge excitations are completely independent of the solver. It is the locally exact nature of the vertex in DMFT in the paramagnetic phase which is the key for these in-gap charge excitations.

## Data availability

All input/output data files that are relevant to run calculations and reproduce all relevant results from the paper are available here on https://doi.org/10.5281/zenodo.8077373 and https://doi.org/10.6084/m9.figshare.23590236. All the input file structures and the command lines to launch calculations are rigorously explained in the tutorials available on the Questaal webpage https://www.questaal.org/get/. Additionally, the same data repository is uploaded with figshare. Any additional request should be made directly to S.A.

## Code availability

The source codes for LDA, QS*GW,* and QS*GŴ* are available from https://www.questaal.org/get/under the terms of the AGPLv3 license.

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

## Acknowledgements

M.I.K., A.I.L., and S.A. are supported by the ERC Synergy Grant, project 854843 FASTCORR (Ultrafast dynamics of correlated electrons in solids). M.vS., D.P., and S.A. (in the later stages of the work) were supported by the Computational Chemical Sciences program within the Office of Basic Energy Sciences, U.S. Department of Energy under

Contract No. DE-AC36-08GO28308. This research used resources of the National Energy Research Scientific Computing Center, a DOE Office of Science User Facility supported by the Office of Science of the U.S. Department of Energy under Contract No. DE-AC02-05CH11231 using NERSC award BES-ERCAP0021783. S.A. and M.I.K. acknowledge PRACE for awarding us access to Irene-Rome hosted by TGCC, France and Juwels Booster and Cluster, Germany. This work was also partly carried out on the Dutch national e-infrastructure with the support of SURF Cooperative.

## Author contributions

M.I.K. and A.I.L. conceived the main theme of the work. S.A. conceived and designed the overall structure of the project. S.A., M.vS., and D.P. have carried out the calculations. C.W. has contributed the ED codes. All authors have contributed to the writing of the paper and the analysis of the data.

## Competing interests

The authors declare no competing interests.
