## [Peer Review File · Nature Communications]

Reviewers' comments:

Reviewer #1 (Remarks to the Author):

In this manuscript the authors conduct an ab initio study of the optical properties of NiO and MnF₂ focusing on excitonic effects, which are responsible for the color of these compounds.

They compare two many-body Green's function theories involving different treatments of the electron-hole vertex, that is the QSGW method developed in Ref. [15] and QSGW+DMFT. The main original result of this work is that both methods applied to the paramagnetic phase of NiO yield a subgap excitonic peak in the correct frequency range to explain the green color of the material. However, only QSGW+DMFT is able to reproduce the pink color of MnF₂, as it also accounts for necessary spin-flip transitions.

My opinion is that the obtained results might be of interest, but are not sufficiently relevant to grant publication in a high impact journal like Nature communications. However, the manuscript has major issues, whereby I may have misinterpreted some aspects and their relevance.

Except for the introduction and the concluding paragraph, the text is carelessly written and in some parts unintelligible. Especially in the Supplementary Information, figures are confusing and not properly explained in the captions. Also it occurs that some significant results are discussed and presented in a way they appear to be new, but are instead taken from an other preprint from the same and other authors. For these reasons, in its present form the manuscript is not suitable for publication in any scientific journal.

Below is an incomplete list of the issues that should be addressed by the authors:

- 1) Refs. [15] and [61] coincide. Here, some of the authors and others had introduced the QSGW method employed in the present manuscript and showed first results on NiO. It appears that the QSGW dielectric function for NiO, which should be one of the main results of the manuscript, was already calculated in that reference. However, the curve for $\text{Im } \epsilon$ displayed in Ref. [15] (pink curve of Fig. 5) and that shown in Supplementary Fig. (SF) 3, mid panel, of the present manuscript look very different. The authors should explain why, and make it clear if the result is new or already known. Also the experimental data used to compare the theoretical curve seem to differ from those of Powell and Spicer, which are cited in the main text, but not in the label of SF3.
- 2) It is not clear how the limiting value of 2.8 eV for optical absorption in NiO has been estimated by the authors. The authors refer to the Supplement, which, however, does not contain any discussion on this point.
- 3) The entire paragraph at page 5 from "A previous GW+BSE calculation..." to the end of the page is not comprehensible and should be reformulated. Same for the paragraph at page 8 "The 1.7eV peak scales with Hund's coupling J....are correctly incorporated within a paramagnetic calculation".
- 5) In the supplementary figures the authors use different acronyms when referring to the same method, which is very confusing to the reader. E.g., the QSGW method introduced in the main text is renamed as BSE in SF1, QSGW+BSE in SF3 and possibly QSGW in SF2. I suggest adopting the same nomenclature of Ref. [15], that is use QSGW if BSE is employed to compute W , and QSGW+BSE if BSE is also used to compute optical properties.
- 6) At page 6 the authors state:
"Most intriguingly, we observe a clear peak in paramagnetic QSGW optical absorption spectrum at ~1.5 eV. The matrix elements between up- and down- spins do not vanish in the paramagnetic state, making this peak optically bright (see Fig. 2)."
It is not clear to which panel of Fig. 2 this statement refers. Possibly the authors mean Fig. 3a.
- 7) Fig.2 is titled:
"Irrelevance of spin-orbit coupling in determining the collective spin and charge excitations in NiO."
This title poorly describes the content of the figure, which is referred to throughout the paper in very

different contexts.

8) In the text eV are often used as a measure of the frequency when discussing graphs where frequencies are in nm and vice versa. I suggest using double axes in the figures.

9) To help the reader, when citing the Supplementary Information the authors should refer to specific items.

In summary, the results of this paper do not appear to be relevant enough to justify publication in a high impact journal as Nature communications. However, the presentation is presently so poor and unclear that a proper review can not be made. The authors should carefully rewrite most of the body of the paper, revise the figures and separate in a clear way novel and former results.

Reviewer #2 (Remarks to the Author):

The manuscript by Acharya et al. examines the photo response of two strongly coupled insulators--- NiO and MnF₂. The former has a pale green color, which can be properly simulated by using a DFT+GW approximation, while the color of MnF₂ requires additional results available only via a DMFT approach.

I find this a very interesting paper and I think it has quite interesting results, really pushing the envelope for what can be done with ab initio types of calculations. I think when modified, it is likely to be appropriate for publication in Nature Communications.

I do have some comments that I think will improve the paper.

(1) I found the paper to be rather short on explaining the details of what is going on. The paper is full of a lot of facts about the calculations, but less so on the synthesis of the results. I think this can be fixed by providing a deeper discussion of why the results come out as they do.

(2) I don't perform these kinds of calculations myself, but I was not clear that enough details have been provided so that someone else could reproduce these results. So, providing enough details to allow this to be done is also important.

(3) Finally, I understand computing the optical response is tricky and sensitive to the distribution of excitations allowed within the optical frequencies, but I was not clear how they took their results and inferred what the actual color was of their systems. I think this needs to be elaborated on further as well, so it can be clear just how they achieve the conclusions they have.

I believe this is an interesting paper with important results and should be published after these issues are properly dealt with.

Reviewer #3 (Remarks to the Author):

The present manuscript addresses the important question of the in-gap optical absorption of two strongly correlated transition-metal insulators, NiO and MnF₂. The authors bring together complementary expertise from the field of many-body perturbation theory (MBPT) and dynamical mean-field theory (DMFT) to tackle the problem both from an ab-initio point of view and from correlated calculations of atomic multiplet structures. In this way, they contribute to reconcile the perspectives from these two competing communities.

The manuscript is well written. Especially the introduction is also clear to non-experts in the field. In

the sections on GW+BSE and DMFT, the amount jargon used in the discussion should be reduced for the sake of clarity and relevance to a broader audience. The manuscript provides important new insight in a long-standing problem. Provided that the issues stated below are thoroughly addressed and can be resolved, it may be suitable for publication in Nature Communications:

1. The introduction is very clear. It comes with an enormous amount of references. However, many of them seem completely off-topic (e.g. the essentially unrelated literature on superconductivity), or highlight only the authors' own contributions to this wide field (e.g. the references on Gw and BSE). Overall, this leaves the impression of notorious self-citation. The authors should provide a more balanced overview of relevant literature.

2. The authors claim that the unfolded paramagnetic bandstructure of NiO strongly resembles the antiferromagnetic band structure. In particular, this touches the problem whether the pure existence of local moments or their long-range order is responsible for the observed electronic structure. This is important for concluding the debate whether the electronic structures of paramagnetic and antiferromagnetic NiO can be considered largely identical or not. However, the plots in the Supp. Mat. do not allow for this conclusion except for the size of the band gap, as the band structures are too involved and shown in separate panels. The authors should do a more detailed comparison, either by unfolding the paramagnetic bandstructure or by folding the antiferromagnetic one and plotting them on top of each other.

3. The authors show that the 1.5 eV excitonic peak in NiO is optically bright in the paramagnetic phase, but not in the antiferromagnetic phase, due to its spin-flip nature. At room temperature, NiO is antiferromagnetic (Neel temperature: 523 K). Why should the 1.5 eV peak then occur in experimental room-temperature absorption spectra, when spin disorder and not SOC or pd hybridization is the reason for its brightness?

4. The intensities in the plots of the susceptibility are given in arbitrary units. The type of calculations the authors performed should yield quantitatively correct intensities. Does the calculated in-gap absorption match in strength with the experimental data or are only the peak positions in agreement? This is an important question for identifying the correct physical mechanism behind the optically weak in-gap transitions and urgently needs to be addressed by the authors.

5. The experimental dielectric function of NiO, extracted by the authors from Powell, Spicer (1970) (Ref. [63]) and shown in Figs. 3 and 4 of the Supp. Mat., has a completely different intensity than in the original publication. The peak height of the imaginary part culminates at about 7 in the original work, here it is downscaled to about 1.2. The authors state that their "computed imaginary part of macroscopic dielectric response ϵ_2 agrees remarkably well [59] with the experimental data from Powell and Spicer [63] over the full frequency range." This is actually not true in view of the rescaling of the experimental curve.

6. Regarding the convergence of the screening function: I understand that the authors used a total of 32 bands to diagonalize the BSE Hamiltonian of NiO. How many bands were included in the evaluation of the screening? 32 bands here would probably be by far too little to converge also the dielectric constant with its contributions from high-energy transitions?

Manuscript preparation issues:

Fig. 1 top: Why do the authors not use the calculated colors of the compounds as given in the Supp. Mat. here, as this is the main result of their work?

Fig. 1 top: The picture of the structure of MnF₂ looks like rocksalt to me. I may be mistaken due to the small size of the picture, but is this really tetragonal MnF₂?

Fig. 1 bottom: From the sketch, the electrons that are excited seem random to me. Are the energy levels represented here to be considered as degenerate as in a ligand-field model of NiO? For MnF₂, the atomic ligand-field model has less degeneracy. Why do exactly the depicted excitations occur? Can the excitations be linked to energy differences?

Fig. 2b: Why are the frequencies negative now?

Fig. 3: For easier comparison with the other spectra, an energy axis would be good here.

We thank the referees for positive assessment of our work. It appears that the criticisms are largely around the presentational quality of some figures in the main text and lack of discussion and coherence of the contents in the supplemental materials. Separate from these, it appears that we could not convince the first referee on the novelty of our work, which as we believe, is in part down to some gap in communications, and in part being distracted by our \widehat{QSGW} dielectric function in NiO, which had been presented in the supplemental material and is now removed.

We thoroughly revised the paper to address all the criticisms/queries from the referees and include new figures and detailed discussions to clarify on several aspects, most importantly, the novelties of our work. At the end, we also include a list for the important revisions in the present version.

Comments/suggestions by the referees helped us improve the presentational quality of our paper significantly and also lead us to spell out clearly some of the fundamental aspects of our understanding of dark and bright excitons in some of the most strongly correlated antiferromagnetic solid state systems. As we show explicitly, our ability to compute excitons, both eigenvalues and eigenfunctions and their spin components, is not only novel but is the right step forward in the direction of studies involving excitons in real materials, particularly, at this crucial juncture in time, where excitons, their orbital and spin components, and their tunability are supposed to shape the future of application of quantum technologies in computation, telecommunications and sensing.

Reply to the reports from Reviewer1

In this manuscript the authors conduct an ab initio study of the optical properties of NiO and MnF2 focusing on excitonic effects, which are responsible for the color of these compounds. They compare two many-body Green's function theories involving different treatments of the electron-hole vertex, that is the \widehat{QSGW} method developed in Ref. [15] and $\widehat{QSGW}+DMFT$. The main original result of this work is that both methods applied to the paramagnetic phase of NiO yield a subgap excitonic peak in the correct frequency range to explain the green color of the material. However, only $\widehat{QSGW}+DMFT$ is able to reproduce the pink color of MnF2, as it also accounts for necessary spin-flip transitions.

My opinion is that the obtained results might be of interest, but are not sufficiently relevant to grant publication in a high impact journal like Nature communications. However, the manuscript has major issues, whereby I may have misinterpreted some aspects and their relevance. Except for the introduction and the concluding paragraph, the text is carelessly written and in some parts unintelligible. Especially in the Supplementary Information, figures are confusing and not properly explained in the captions. Also it occurs that some significant results are discussed and presented in a way they appear to be new, but are instead taken from an other preprint from the same and other authors. For these reasons, in its present form the manuscript is not suitable for publication in any scientific journal.

We revise the paper significantly. We believe, the referee is suggesting the presentation of the figure on macroscopic dielectric response of NiO in Fig 3 in supplemental materials. This figure was included for completeness and it is largely unrelated to the essential results of the work (it was consigned to the supplemental material) namely, what makes NiO green. This depends on the ability to produce deep excitons and their effect on the dielectric response. The

text has been revised to discuss the character of the deep excitons at length. To the best of our knowledge, this is the first time these excitons have been discussed; it is the first time a true *ab initio* method for periodic systems has been applied to it, and it is the first time excitons in two orthogonal approaches (perturbation theory vs DMFT) have been compared for any system; it is the first time the role of spin disorder has been introduced as symmetry-lowering mechanism affecting the brightness of excitons, and the first time, to our knowledge, the spin and charge response functions have been compared in a wide-gap system to show how they depend differently on the correlation strengths; and finally, the ability to predict the color in MnF_2 provided the theory includes spin-flip diagrams in the charge response, is new. All of these contributions are novel, and also highly nontrivial. That the nonlocal perturbative and local non-perturbative approaches yield so similar results is by itself a significant result. It is difficult — and rare — to apply DMFT to color centres in wide-gap materials, not least because of challenges with analytic continuation. The work closest to our own was published by Rödl and Bechstedt (PHYSICAL REVIEW B 86, 235122 (2012)). Their computed BSE dielectric response shows reasonable agreement with the classic experimental data from Powell and Spicer. However, their work is semi-empirical: they add Hubbard U and J , and a scissors shift as well, and further scale the result because their computed ϵ_∞ is poor; also their work misses the excitons responsible for color. The only successful approach, in the literature, for $d-d$ transitions in NiO, uses simple multiplet ligand-field theory combined with exact diagonalization of small $[\text{NiO}_6]$ cluster (M. W. Haverkort, et al. Phys. Rev. B 85, 165113 (2012)). We use a true *ab initio* approach, which yields a good $\epsilon(\omega)$ (including ϵ_∞) and crucially for the present work — yields a good description of the color of NiO.

We believe, we are able to sufficiently clarify these points in the present text, and they also better address the following questions from the referee, which we answer individually. We have taken out the figure from the supplemental material and have added it here. This, we believe, is important to address the fact that this particular figure has no bearings on the essential results of our work.

1. Refs. [15] and [61] coincide. Here, some of the authors and others had introduced the method employed in the present manuscript and showed first results on NiO. It appears that the QSGW dielectric function for NiO, which should be one of the main results of the manuscript, was already calculated in that reference. However, the curve for $\text{Im}\epsilon$ displayed in Ref. [15] (pink curve of Fig. 5) and that shown in Supplementary Fig. (SF) 3, mid panel, of the present manuscript look very different. The authors should explain why, and make it clear if the result is new or already known. Also the experimental data used to compare the theoretical curve seem to differ from those of Powell and Spicer, which are cited in the main text, but not in the label of SF3.

As we now have clarified, the figure that was presented in the supplemental material in the initial version is not relevant for the main theme/results of our paper. However, we kept the figure in the supplemental material since it was, as we thought, important for the benchmarking purpose against available experimental results at those energy scales (> 3.0 eV).

The excitons of our interest are in the energy range 1.5-3.0 eV and are presented in the Figure 3 in the main paper and for that the energy scale there

are no optical absorption peaks in the data from Powell and Spicer. However, there are peaks observed in the IXS data from G. Ghiringhelli et al. between 1.5 and 3.0 eV and we reproduce them in the main text to benchmark against our computed excitons. The same peaks, as we have found from calculations and observed in IXS, were also observed in a classic work on optical emission spectroscopy by Propach and Reinen (1978). As we have discussed in detail, this is extremely important. IXS is insensitive to the optical matrix element effects and optical emission spectroscopy is sensitive to non-radiative coupling of e-h processes. Both the experimental approaches, hence, 'see' these deep lying excitons, which was not detected by Powell and Spicer in their absorption spectroscopy. Within our theoretical approach we pick up all such excitons (albeit with small optical matrix elements), because our approaches contain the essential ingredients of the interaction vertex. How bright these excitons are, depend on symmetry lowering mechanisms but their existence does not. This is of utmost importance and is often not appreciated by the community. We believe we have been able to sufficiently emphasise this in the revised version.

2) It is not clear how the limiting value of 2.8 eV for optical absorption in NiO has been estimated by the authors. The authors refer to the Supplement, which, however, does not contain any discussion on this point.

We find an excitonic eigenvalue of 2.8 eV when the BSE e-h hamiltonian is diagonalised in the paramagnetic phase. We also clarify any ambiguities related to this aspect in the main text. Further, we have added a figure to the supplemental materials precisely showing the convergence of the excitonic peak at 2.8 eV in paramagnetic NiO.

3) The entire paragraph at page 5 from "A previous GW+BSE calculation..." to the end of the page is not comprehensible and should be reformulated. Same for the paragraph at page 8 "The 1.7eV peak scales with Hund's coupling J...are correctly incorporated within a paramagnetic calculation".

We have rewritten the text.

5) In the supplementary figures the authors use different acronyms when referring to the same method, which is very confusing to the reader. E.g., the $QSG\widehat{W}$ method introduced in the main text is renamed as BSE in SF1, QSGW+BSE in SF3 and possibly QSGW in SF2. I suggest adopting the same nomenclature of Ref. [15], that is use $QSG\widehat{W}$ if BSE is employed to compute W, and $QSG\widehat{W}+BSE$ if BSE is also used to compute optical properties.

We remove all inconsistencies in the definition and in naming the methods.

6) At page 6 the authors state: 'Most intriguingly, we observe a clear peak in paramagnetic $QSG\widehat{W}$ optical absorption spectrum at ~ 1.5 eV. The matrix elements between up- and down- spins do not vanish, making this peak optically bright (see Fig. 2).' It is not clear to which panel of Fig. 2 this statement refers. Possibly the authors mean Fig. 3a

We have corrected all inconsistencies in referencing the figures all through the text. SO coupling can make the exciton bright. As we showed that spin disorder can also., which we modeled by the paramagnetic state though it occurs at any temperature.

7) Fig.2 is titled: “Irrelevance of spin-orbit coupling in determining the collective spin and charge excitations in NiO.” This title poorly describes the content of the figure, which is referred to throughout the paper in very different contexts. 8) In the text eV are often used as a measure of the frequency when discussing graphs where frequencies are in nm and vice versa. I suggest using double axes in the figures. 9) To help the reader, when citing the Supplementary Information the authors should refer to specific items.

This was somewhat of a misstatement. The intention was to note that spin-orbit coupling has little effect on the excitons themselves. However it can affect the brightness; it is one of a number of possible mechanisms. We have revised the text.

In summary, the results of this paper do not appear to be relevant enough to justify publication in a high impact journal as Nature communications. However, the presentation is presently so poor and unclear that a proper review can not be made. The authors should carefully rewrite most of the body of the paper, revise the figures and separate in a clear way novel and former results.

We believe we could revise the paper in the desired manner that main theme of the work and its novelties are clearly presented in the revised version. We noted above the several significant and nontrivial points of novelty. In the wider view, this work presents for the first time a true ab initio theory that does not rely on the ligand-field approximation or model parameters, that can fully account for colors in such wide-gap insulators.

Reply to the reports from Reviewer2

The manuscript by Acharya et al. examines the photo response of two strongly coupled insulators—NiO and MnF₂. The former has a pale green color, which can be properly simulated by using a DFT+GW approximation, while the color of MnF₂ requires additional results available only via a DMFT approach. I find this a very interesting paper and I think it has quite interesting results, really pushing the envelope for what can be done with ab initio types of calculations. I think when modified, it is likely to be appropriate for publication in Nature Communications.

We thank the referee for the positive assessment of our work and for his/her recommendation for publication of our work in Nature Communications.

I do have some comments that I think will improve the paper.

(1) I found the paper to be rather short on explaining the details of what is going on. The paper is full of a lot of facts about the calculations, but less so on the synthesis of the results. I think this can be fixed by providing a deeper discussion of why the results come out as they do.

We believe we have addressed this issue now. We have added several paragraphs to discuss the important aspects of the work.

(2) I don't perform these kinds of calculations myself, but I was not clear that enough details have been provided so that someone else could reproduce these results. So, providing enough details to allow this to be done is also important.

We provide a data repository with all the input and output files necessary to reproduce the results.

(3) Finally, I understand computing the optical response is tricky and sensitive to the distribution of excitations allowed within the optical frequencies, but I was not clear how they took their results and inferred what the actual color was of their systems. I think this needs to be elaborated on further as well, so it can be clear just how they achieve the conclusions they have.

We have addressed this at length in the revised version.

I believe this is an interesting paper with important results and should be published after these issues are properly dealt with.

We thank the referee again for this.

Reply to the reports from Reviewer3

The present manuscript addresses the important question of the in-gap optical absorption of two strongly correlated transition-metal insulators, NiO and MnF₂. The authors bring together complementary expertise from the field of many-body perturbation theory (MBPT) and dynamical mean-field theory (DMFT) to tackle the problem both from an ab-initio point of view and from correlated calculations of atomic multiplet structures. In this way, they contribute to reconcile the perspectives from these two competing communities. The manuscript is well written. Especially the introduction is also clear to non-experts in the field. In the sections on GW+BSE and DMFT, the amount of jargon used in the discussion should be reduced for the sake of clarity and relevance to a broader audience. The manuscript provides important new insight in a long-standing problem. Provided that the issues stated below are thoroughly addressed and can be resolved, it may be suitable for publication in Nature Communications

We thank the referee for his/her positive assessment of our work. And we believe with the revisions, we are able to satisfactorily address all the queries/criticisms.

1. The introduction is very clear. It comes with an enormous amount of references. However, many of them seem completely off-topic (e.g. the essentially unrelated literature on superconductivity), or highlight only the authors' own contributions to this wide field (e.g. the references on GW and BSE). Overall, this leaves the impression of notorious self-citation. The authors should provide a more balanced overview of relevant literature.

We have removed all citations, about 25 of them, (self- or otherwise) that are not related to the computation of collective charge responses in strongly correlated systems. We have added references for different experiments on optical absorption/emission spectra on NiO to add value to the paper. We also cite some previous theoretical works on NiO, to better establish what is novel in our work.

2. The authors claim that the unfolded paramagnetic bandstructure of NiO strongly resembles the antiferromagnetic band structure. In particular, this touches the problem whether the pure existence of local moments or their long-range order is responsible for the observed electronic structure. This is important for concluding the debate whether the electronic structures of paramagnetic and antiferromagnetic NiO can be considered largely identical or not. However, the plots in the Supp. Mat. do not allow for this conclusion except for the size of the band gap, as the band structures are too involved and shown in separate panels. The authors should do a more detailed comparison, either by unfolding the paramagnetic bandstructure or by folding the antiferromagnetic one and plotting them on top of each other.

We fully agree with the referee. We perform paramagnetic QSGW calculations to explore the effect of paramagnetism on driving dark excitons bright. The referee is entirely correct in stating that we can only compare the band gap changes between AFM and paramagnetic phase from our presented data. We revise the text to clearly bring out this aspect. Unfolding the paramagnetic band structure is a significant undertaking since our method is not a simple plane wave method. However, it is not difficult to make the AFM band structure in the PM unit cell, which we have done and included the new figures in the supplemental materials.

3. The authors show that the 1.5 eV excitonic peak in NiO is optically bright in the paramagnetic phase, but not in the antiferromagnetic phase, due to its spin-flip nature. At room temperature, NiO is antiferromagnetic (Neel temperature: 523 K). Why should the 1.5 eV peak then occur in experimental room-temperature absorption spectra, when spin disorder and not SOC or pd hybridisation is the reason for its brightness?

The 1.6 eV exciton is always present in NiO. In real world, whether it is above Neel temperature or below, NiO has some finite spin orbit coupling that can possibly make this peak brighter. However, a number of other symmetry-lowering mechanisms (or combinations of them) can accomplish this. Here we show that the effect of spin-disordering (paramagnetism) on the spin-dependent optical matrix elements is similar to that of including SO within a fully ordered AFM phase. In a future work, we will compare the relative contributions of different symmetry-lowering mechanisms on brightness, e.g. SO coupling, pd hybridization, lattice vibrations, spin disorder.

4. The intensities in the plots of the susceptibility are given in arbitrary units. The type of calculations the authors performed should yield quantitatively correct intensities. Does the calculated in-gap absorption match in strength with the experimental data or are only the peak positions in agreement? This is an important question for identifying the correct physical mechanism behind the optically weak in-gap transitions and urgently needs to be addressed by the authors.

We are computing optical absorption spectra, while the experiments where these peaks are observed are either, optical emission spectra or IXS. Powell and Spicer fail to observe the peaks in their absorption spectra as well. IXS is not sensitive to optical matrix element effects while emission spectra has significant non-radiative exciton-boson couplings. So it is only natural that the intensities of the exciton peaks from our theory (or for that matter from the experiments of Powell and Spicer) should not agree with the IXS or emission spectra. This is one primary reason for including the new figure, Fig.2, in the main text to discuss the oscillator strengths of different deep lying excitons in a range of systems. A proper theory for many-body interaction should contain all such excitons (at the correct energies), while their intensities depend on multiple factors and boson-boson couplings of different origins.

Further, we agree with the referee and this is indeed our goal; to be able to compute the intensities and broadening of individual excitonic peaks in real materials. However, a full description of the intensity also requires a proper description of all possible mechanisms that cause finite lifetime. These two aspects are related, as the excitons couple with available other bosons in the systems, their intensities and lifetimes change. Additionally, spin-orbit coupling or p-d hybridisation also can change the intensities of these excitonic peaks. This is a work in progress and we are not there yet. That being said, the mechanisms that contribute to the lifetime and/or oscillator strengths of these excitonic peaks, is distinct from the mechanism that generates the excitons in the first place, which is a primary focus of our work. For these excitons that are essentially ‘dipole’ forbidden and exist in the atomic limit, almost across the entire body of literature, their existence is attributed to some symmetry lowering mechanism. What we show in our work that there exists a large gap in our knowledge about the existence of these excitons outside the molecular approximations found in

the quantum-chemical literature; our ability to compute vertex functions within newly developed many-body approaches suggest that most of these excitons exist in the materials, purely through electronic many body interactions. We include a new figure and lengthy discussions around the oscillator strengths of excitons from different kinds of sp, dd systems to address this entire aspect.

5. The experimental dielectric function of NiO, extracted by the authors from Powell, Spicer (1970) (Ref. [63]) and shown in Figs. 3 and 4 of the Supp. Mat., has a completely different intensity than in the original publication. The peak height of the imaginary part culminates at about 7 in the original work, here it is downscaled to about 1.2. The authors state that their "computed imaginary part of macroscopic dielectric response ϵ_2 agrees remarkably well [59] with the experimental data from Powell and Spicer [63] over the full frequency range." This is actually not true in view of the rescaling of the experimental curve.

Yes, this is true : we inadvertently submitted the wrong figure in our original submission. That figure was adapted from prior work, where the scale is correct. In this revision we have removed this figure from the supplemental material since it does not add any value to the essential discussions in the main text and also is a duplication of the already published figure by us in a separate journal, moreover it creates confusion (see comments by the first referee). The figure is supplied with this response.

6. Regarding the convergence of the screening function: I understand that the authors used a total of 32 bands to diagonalise the BSE Hamiltonian of NiO. How many bands were included in the evaluation of the screening? 32 bands here would probably be by far too little to converge also the dielectric constant with its contributions from high-energy transitions?

The code is written so that the RPA part of $\epsilon(\omega)$ is calculated first, and the difference between the BSE and RPA parts added to it. At the RPA level, $\epsilon(\omega)$ is computed with the entire basis, so the number of states needed for the BSE augmentation is relatively small. In the supplemental material, we show the convergence plot for excitonic eigenvalues in paramagnetic NiO with number of valence and conduction states included in BSE. For the paramagnetic supercell, we included 96 valence and 96 conduction states.

Manuscript preparation issues: Fig. 1 top: Why do the authors not use the calculated colours of the compounds as given in the Supp. Mat. here, as this is the main result of their work?

Such pictures are produced using a java applet available with Quantum Espresso. They pictorially represents the same amount of information, that we believe, we could physically justify sufficiently in the text. We did not feel the need to keep this figure in the paper and have removed it from SM too, in the revised version. If the referee considers this to be important and unavoidable, we will be happy to include it in the paper.

Fig. 1 top: The picture of the structure of MnF2 looks like rocksalt to me. I may be mistaken due to the small size of the picture, but is this really tetragonal MnF2?

The referee is right. We have corrected the presentation of MnF₂ crystal structure.

Fig. 1 bottom: From the sketch, the electrons that are excited seem random to me. Are the energy levels represented here to be considered as degenerate as in a ligand-field model of NiO? For MnF₂, the atomic ligand-field model has less degeneracy. Why do exactly the depicted excitations occur? Can the excitations be linked to energy differences?

This is an extremely interesting question. The excitations are multiplet transitions involving 5/2, 3/2 and 1/2 spin states of Mn atom. Within a fully dynamic vertex (from DMFT) all such transitions and combinations of them are allowed.

In NiO, we have an exciton at 1.6 eV in both the ordered AFM phase and the PM phase, from QSGW. In the AFM phase, there is no spin mixing in the BSE for the electron and hole. Since, e_g is half filled, such triplet transition can take place within AFM QSGW, if a down spin from fully filled t_{2g} states jump to half-filled e_g state. This is allowed in the AFM BSE. Note that, there is crystal field splitting between t_{2g} and e_g states, however, the vertex allows for such transitions.

Degeneracy of the ligand field model is often thought to be precursor for the atomic multiplet transition, however, the NiO case explicitly shows that it does not have to be. If we could compute vertex functions with all its essential ingredients, it should allow for multiplet transitions even when degeneracy is lacking. This is exactly what happens in MnF₂.

Separate from this, we have presented only a schematic diagram of the atomic levels. We have revised the figure to bring out the information of degeneracy in NiO and non-degeneracy in MnF₂. We thank the referee for this suggestion.

Fig. 2b: Why are the frequencies negative now?

The peak-structure is the same in both the positive and negative bosonic frequency grids. We have revised the figure now with the positive part of the grid only.

Fig. 3: For easier comparison with the other spectra, an energy axis would be good here.

We have revised the figure accordingly.

Below we list the important changes made in our revised version of the paper. All changes to the text are in color red.

- The text is thoroughly revised.
- MnF_2 crystal is now correctly presented in Fig. 1.
- Degeneracies of the d states are correctly presented in Fig. 1.
- A complete new figure, Fig 2, is included.
- Fig. 3 and Fig. 4 are updated.
- Online repository is made available with all relevant input and output files for reproducing all results/figures from the paper.
- Several sections and figures are added in SM.
- New figure for eigenvalue convergence is added in SM.
- AFM and PM band structures are compared in the same supercell configuration.
- The NiO dielectric response figure is removed from SM.
- All suggestions by the referees that are related to the presentation of figures in main text and SM, are included.
- Several experimentally and theoretically relevant references are included (>10).
- All citations that are concerned with the role of vertex in 3d system, but not relevant for the excitons, are removed.

REVIEWERS' COMMENTS

Reviewer #1 (Remarks to the Author):

The authors have addressed all my comments and rewritten large sections of the manuscript improving the clarity of the presentation. I appreciate that the results are of interest. However, I still find the paper rather incremental. To summarize:

1) The QSGW method was proposed in Ref. [37] and has been already applied by the authors to study deep excitons in CrX₃ [25]. In the present manuscript the authors state that "In analogy with CrX₃, in NiO, the Frenkel excitons with atomic $d_{8\uparrow} \rightarrow d_{8\uparrow}$ transitions should still be picked up in many-body perturbative framework since an essentially atomic triplet exciton can emerge without requiring a spin flip". This expected result is indeed confirmed by showing that QSGW captures the 1.6 eV Frenkel exciton required to reproduce the green color of NiO.

2) The fact that both QSGW and DMFT can reproduce the band gap of paramagnetic NiO (which demonstrates the marginal role played by dynamical spin fluctuations) is definitely of interest, but confirms a result already obtained in Ref. [51].

3) As for the spin-disorder as an alternative/concurrent mechanism to make the dark excitons bright, its relevance should be assessed against the spin-orbit coupling by including the latter in the calculations.

In conclusion, I believe that the level of novelty and advance is not sufficient to grant publication in a high impact factor journal as Nature Communications. Also, the paper remains quite technical and in my opinion addressed to a more specialized audience.

Reviewer #2 (Remarks to the Author):

This manuscript has had a significant rewrite to it. I was favorably disposed to the first draft under the assumption that a better description of how the calculations are performed was included. It has been. The modifications have also improved the manuscript in many other places as well. I am impressed with the level of sophistication of the work and feel it is now ready for acceptance to the journal.

We thank the referees for positive assessment of our work and for recommending our work for publication. We thank all the referees from both the first and second phases of the referrals since their comments/suggestions helped us improve the quality of presentation of our paper significantly and bring more clarity in stating some of the fundamental aspects of our work.

Reply to the reports from Reviewer1

The authors have addressed all my comments and rewritten large sections of the manuscript improving the clarity of the presentation. I appreciate that the results are of interest. However, I still find the paper rather incremental. To summarize:

2) The fact that both $QSG\widehat{W}$ and DMFT can reproduce the band gap of paramagnetic NiO (which demonstrates the marginal role played by dynamical spin fluctuations) is definitely of interest, but confirms a result already obtained in Ref. [51].

This is an understatement. The band gap from two foundationally different theories (one with locally exact diagrams and the other with perturbative non-local diagrams) can be made to come out the same using several free parameters. With sufficient tweaking of parameters; for example, by tweaking U_{dd} , U_{pd} and U_{pp} , even LDA+U can be made to produce a 4 eV band gap in NiO (note we don't have a single free parameter in our $QSG\widehat{W}$ theory). However, this does not imply that any theory can capture the most important features of the band structure; most importantly the p-d alignments at different energies and at different \mathbf{q} points. Optical conductivity, and even more intriguingly the color of a material, is probably the most convincing, and, at the same time, the most challenging check for the quality of electronic band structure of a material (for that reason the paper discusses elemental copper, gold, silver in the introduction.)

However, for NiO and MnF_2 , even that argument fails. A theory that produces the correct single-particle band structure for them will fail entirely in producing the colors for these materials unless the two-particle vertex functions can be calculated reliably. We believe we have sufficiently addressed the concern expressed by the referee by being able to build theories that can do both without the use of a single free parameter. It probably would not be fair to reduce something this important to a mere comparison of band gaps from different kinds of approaches. Moreover, the reference [51] does not shed any light on whether the dynamical vertex is not important for the physics in NiO and also can not address whether dynamical vertex would be irrelevant for a system with different valence and multiplet structure, for example a system like MnF_2 .

To put it in the larger perspective, the high barrier to computing vertex functions in different channels in real materials is what prevents us from gaining systematic understanding into the crucial mechanisms two-particle properties in real materials, the optical response being one and unconventional superconductivity another. We believe we have provided the most remarkable examples of the importance of computing both 'necessary and sufficient' vertex functions fully ab-initio in the charge channels for the relevant systems here. It opens a way to the next level in electronic structure calculations and, in our opinion, should not be considered as something not significant enough. Transition from density functional theory (DFT) to Green's function functionals such as GW and, much later, DFT+DMFT, was a breakthrough. We believe that the leap in

our computational ability from single-particle Green's functions to two-particle Green's functions including vertices in practical calculations is a breakthrough, perhaps, of comparable importance.

1. The QSGW method was proposed in Ref. [37] and has been already applied by the authors to study deep excitons in CrX₃ [25]. In the present manuscript the authors state that "In analogy with CrX₃, in NiO, the Frenkel excitons with atomic d8→d8 transitions should still be picked up in many-body perturbative framework since an essentially atomic triplet exciton can emerge without requiring a spin flip". This expected result is indeed confirmed by showing that QSGW captures the 1.6 eV Frenkel exciton required to reproduce the green color of NiO.

We do not stop at only computing and explaining the need of high quality vertex functions that explain the colors of these systems, but we go on to generalize the formulation and establish a language for the vertex mediated optical transitions that both chemists and physicists would equally appreciate, by connecting the necessary minimum ingredients of a theory in describing the multiplet transitions from various systems. Different multiplet configurations may need principally different approaches to computing vertex functions. This is no mean feat. It is only in this context that we use the examples of TMD's and CrX₃. We believe it is not fair to tag such observations as 'incremental' or 'expected results'. If it was so, we wonder why a large fraction for the community would keep attributing the existence of most such excitons to spin-orbit coupling or couplings with phonons. It is primarily for these reasons that we published the works that the referee mentioned, on CrX₃, that addresses some of these foundational questions on existence and tuning of excitons in those systems. If we fail to understand why these excitons exist, we will keep using the 'wrong' knobs to tune them.

3) As for the spin-disorder as an alternative/concurrent mechanism to make the dark excitons bright, its relevance should be assessed against the spin-orbit coupling by including the latter in the calculations.

SOC plays the trivial role of changing the oscillator strength of the 1.6 eV peak by certain amount. However, it does not matter whether we have spin-orbit coupling or not, the exciton in NiO remains dark. We included a table now in the revised supplemental materials with the oscillator strengths of the relevant excitonic absorptions with and without SOC. We also include a figure for optical absorption with and without SOC. However, we would like to stress the fact that the paper already contains an entire section titled 'Different scaling features of peaks in spin and charge' which discusses the irrelevance of spin-orbit coupling in existence of these dark excitons. Most importantly, there exists a detailed discussion around two independent experimental observations of fundamental importance in that same section;

1) recent X-ray measurements by Haverkort et al., which is insensitive to the optical matrix elements, where all these three peaks are observed at almost the same energies as our theory and the classic work by Propach and Reinen, they picked up all the three peaks in the optical emission spectra too.

2) however, Powell and Spicer failed to see these peaks in their optical absorption spectra.

Both optical emission and absorption spectra contain the same degree of spin-orbit coupling that is present in the material. But the absence of the peaks

in absorption spectra and presence of them in the emission spectra suggest that SOC can not be the mechanism that brightens the peaks, but it is most likely the coupling with other radiative/non-radiative mechanisms that can only brighten the peak in the emission spectra. However, an experiment (X-ray) that does not depend on the optical-matrix element effects, can see the peaks. We have added a few sentences now in the main text discussing these points.

Below we list the important changes made in our revised version of the paper. All changes to the text are in color red.

- A figure and table are added to address the impact of SOC on the dark excitons.
- A paragraph is added to the main text for the same.
- Abstract is significantly shortened. It is now about 180 words, shortened from about 300 words as it was before.
- All editorial suggestions are taken into account.